# Seasonal Variability in the Prevalence of DWV Strains in Individual Colonies of European Honeybees in Hawaii

**DOI:** 10.3390/insects15040219

**Published:** 2024-03-23

**Authors:** Zhening Zhang, Ethel M. Villalobos, Scott Nikaido, Stephen J. Martin

**Affiliations:** 1Department of Plant and Environmental Protection Sciences, College of Tropical Agriculture and Human Resources, University of Hawaii, 3050 Maile Way, 310 Gilmore Hall, Honolulu, HI 96822, USA; emv@hawaii.edu (E.M.V.); snikaido@hawaii.edu (S.N.); 2School of Environment and Life Sciences, The University of Salford, Manchester M5 4WT, UK; s.j.martin@salford.ac.uk

**Keywords:** colony-level surveillance, viral load fluctuations, prevalence dynamics, Hawaiian Islands

## Abstract

**Simple Summary:**

Deformed wing virus (DWV) is the most widespread bee virus and one of the most destructive pathogens affecting honeybees. Initially believed to be a honeybee virus with one dominant variant, DWV-A, we now know there are four master variants DWV-A, B (VDV-1), C and D. Since 2010, a shift in prevalence from the originally dominant DWV-A strain towards an increase in DWV-B prevalence has been observed in several countries. The geographical isolation of the bee populations in Hawaii provides a unique opportunity to examine the changes in DWV strain prevalence both at the individual colony and the apiary level. In this study, we present longitudinal data on the prevalence of DWV strains in individual colonies on the Island of Oahu, Hawaii. We found that although DWV-A was the dominant strain in the apiary, individual colonies could now be characterized as a mix of DWV-A and DWV-B and some tested positive for DWV-B only. Finally, DWV-B exhibited seasonal variations in both viral loads and prevalence in the apiary. Our long term DWV monitoring at the individual colony level contributes a unique perspective on the emerging patterns of DWV-B.

**Abstract:**

The most prevalent viral pathogen of honeybees is Deformed Wing Virus (DWV) and its two most widely studied and common master-variants are DWV-A and DWV-B. The prevalence of DWV variants in the UK and in the US is changing, with the prevalence of the DWV-A strain declining and DWV-B increasing over time. In 2012, only DWV-A was detected on the Hawaiian Islands of Oahu. In this study we focused on a colony-level survey of DWV strains in a single apiary and examined the prevalence of DWV variants over the course of two years. In 2018 and 2019, a total of 16 colonies underwent viral testing in January, May, and September. Of those 16 colonies, four were monitored in both 2018 and 2019. Individual colonies showed variability of DWV master variants throughout the sampling period. DWV-A was consistently detected; however, the detection of DWV-B was variable across time in individual colonies. Ultimately, this study demonstrated a seasonal variation in both viral prevalence and load for DWV-B, providing a perspective on the dynamic nature of DWV master variants emerging in Hawaii.

## 1. Introduction

Deformed wing virus (DWV) was first identified in the early 1980s [1]. DWV evolved in association with European honeybees (*Apis mellifera*) [2] and later became associated with their ectoparasite varroa mites (*Varroa destructor*) [3]. DWV is an RNA virus that characteristically exhibits high reproductive and mutation rates. Consequently, it quickly evolves new strains that spread throughout the host population. Initially, two master variants of DWV were described: DWV-A and DWV-B, which are the most widely detected and studied, and are both related to the common disease symptoms [4].

DWV-A was initially the most prevalent bee virus in Europe and the U.S. [5,6] but now has been outcompeted by the DWV-B strain which is now widespread in Germany, U.K., and Italy [7]. By 2016, DWV-B was not only widespread but had become the dominant strain in England and Wales [8]; it is now also the most prevalent DWV variant in Europe [9]. Although DWV-B has not become dominant in the USA, the prevalence of this strain appears to have increased over time; from 2.3% in 2010 [8] to 23% in 2019. Likewise, in Hawaii DWV-A was the original dominant strain while DWV-B was first detected in 2012. However, 10 years later, DWV-B accounts for almost half of the viral load in Hawaii’s colonies [10].

The Hawaiian archipelago is a closed system, lacking the more fluid hive movements between geographically distant locations that occur in the continental USA. Consequently, it provided researchers with a rare opportunity to observe, in real-time, the changes that occured in the viral landscape when the varroa mites were introduced into the mite-naïve Hawaiian honeybee population in 2007. Because of this relatively short history of coevolution between varroa mites and honeybees, Hawaii has played an important role in understanding the evolution of DWV.

In the present study we examined the shift in dominance between DWV strains by tracking individual colonies over time and recording the changes in prevalence, persistence, and relative dominance of DWV variants A and B for each colony. The longitudinal data obtained from the colonies in a single apiary on the island of Oahu highlight the dynamic nature of DWV variants and the potential for fast strain shifting.

## 2. Materials and Methods

### 2.1. Location and Colony Management

Samples were collected from January 2018 to September 2019 at the apiary at the University of Hawaii Waimanalo Agricultural Research Station, Oahu, HI, USA (21.3355261N, −157.714788W). Initially, there were ten healthy colonies of similar size selected in the first-year study. However, due to poor requeening and environmental issues, six colonies were excluded by the end of 2018 because of their small colony population size. To maintain the sample size, six colonies, which were located at the same apiary, were added to this study starting in January 2019. As a result, only four colonies were tracked throughout both years. All study colonies were managed using standard apicultural techniques and all were treated for varroa mites using Apivar^®^ (Veto-pharma, Palaiseau, France) in November 2017, October 2018, and February 2019.

Around 200–300 individual nurse bees per hive were sampled every four months (January, May, and September) and samples were stored at −80 °C. The number and identity of the colonies included in each sampling period are listed in Figure 1.

### 2.2. RNA Extraction

A total of 30 asymptomatic adult bees were selected from each colony’s sample for RNA extraction. Bees that had visibly deformed wings or were infested with varroa mites were not included in the sample. Samples were flash-frozen with liquid nitrogen and ground into a fine powder using a sterilized mortar and pestle. A 60 mg subsample of the crushed powder was transferred to a nuclease-free 1.5 mL centrifuge tube for RNA extraction. Total RNA extraction was conducted using an RNeasy Mini prep kit (QIAGEN, Germantown, MD, USA) according to manufacturer’s conditions and resuspended with 30 μL RNase-free water. RNA concentration was determined using a Nanodrop 2000c (Thermo Fisher Scientific, Waltham, MA, USA), and samples were standardized to 50 ng/µL. An aliquot of 100 ng of extracted RNA was used to synthesize complementary DNA (cDNA) using a Quantitect Reverse Transcriptase Kit (QIAGEN, Germantown, MD, USA) following the manufacturer’s protocols.

### 2.3. RT-qPCR

The cDNA samples were standardized to 50 ng/μL using a Qubit fluorometer (Thermo Fisher Scientific, Waltham, MA, USA) and were quantified for both DWV-A and DWV type B via RT-qPCR using SsoAdvanced Universal Inhibitor-Tolerant SYBR^®^ Green Supermix (Hercules, CA, USA). For analysis, each sample was amplified in duplicate samples with a negative control and positive controls (10^5^ gene copies of standards) in a 10 μL reaction of 2 μL cDNA, 2 μL primer containing 0.5 μM of each primer, 5 μL Supermix and 1 μL RNeasy water. Primers used in this study were from Kevil et al. (2017) [11]: DWV Forward: TACTAG-TGCTGGTTTTCCTTT and reverse primers: DWV-A Reverse: CTCATTAACTGTGTCGTTGAT and DWV-B Reverse: CTCATTAACTGAGTTGTTGTC.

The positive controls used in the qPCR test were amplified and purified from already known positive samples in the same field. The qPCR standard curves were generated using ten-fold serial dilutions of purified DNA amplicons. The qPCR amplification efficiencies of calibration curves were in the range of 96.7% and 93.6% with an R^2^ range of 1 and 0.99 for DWV-A and DWV-B, respectively.

The thermocycler was programmed for enzyme activation at 95 °C for 30 s, followed by 40 cycles of denaturation at 95 °C for 5 s and annealing/extension at 60 °C for 30 s. The threshold cycles (Ct) of each sample were then recorded and compared with standard curves. The analysis was run in duplicate, and the means were used for calculation. Any samples which had a greater than 3 Ct values difference were reanalyzed. The calculation of copy numbers of the template assumes that the average weight of a base pair (bp) is 650 Daltons. The viral loads were expressed in genome equivalents which were calculated using the equation based on Kevel et al. (2017) [11].

The statistical analysis was carried out in JMP14 Pro (SAS Institute, Cary, NC, USA) to compare the viral loads among individual bees and among colonies at the apiary level. Data was not normally distributed and consequently non-parametric tests were utilized. The viral load between different DWV types and infection type (single/double infection of DWV variants) was compared using the Wilcoxon Signed-Rank non-parametric test. Viral loads of DWV-A and B among the three sampling times were compared for each pair using the Wilcoxon method (non-parametric). A Fisher’s Exact test was used to compare DWV infection prevalence between DWV-A and DWV-B. The significance level (α) was set as 0.05. Data were plotted in PowerPoint (Microsoft, Redmond, WA, USA) and Origin 2021 (OriginLab Corporation, Northampton, MA, USA).

## 3. Results

### 3.1. Infection Prevalence of DWV-A and DWV-B

All tested colonies, except for two that tested negative during the first month (Jan 2018), tested positive for either DWV-A, DWV-B, or both. The two colonies that were negative in January 2018 subsequently tested positive for both strains in May-2018. The prevalence rate of DWV-A (93% and 93% in 2018 and 2019, respectively) was higher than DWV-B (47% and 48% in 2018 and 2019, respectively) during both years. The co-occurrence of both DWV strains was common, and 49% of samples that were positive for DWV-A were also positive for DWV-B. The type of infection of each colony across time, whether negative, positive with a single variant, or a mix of both DWV strains is shown in Figure 1.

DWV-A was persistent throughout the year in nearly all samples; in contrast, DWV-B was intermittently detected across different sample periods. However, the infection rate of DWV-B in the apiary was higher in May (70% and 80% in 2018 and 2019, respectively) of this study (Figure 1b). In contrast, the prevalence of single infections by DWV-A in colonies was higher in September for both years (60% and 71% for 2018 and 2019, respectively). The detection of strain B in a sample did not appear to increase or reduce the probability that the same colony would continue to test positive for DWV-B in future samples (Figure 2). Colonies infected with only DWV-A were relatively more likely to revert to DWV-A again (9/17 of occurrences) and the second most likely change was from a single DWV-A infection to a double infection of DWV-A and B (6/17 occurrences). Double strain infections could revert to DWV-A (8/21 occurrences), but not to single DWV-B infection. The appearance of DWV-B as a single strain infection was relatively rare, and the colony status changed to a pure DWV-A strain in the next sampling period. However, when DWV-B was detected as part of a double strain infection (DWV-A and B) in an individual colony, it was likely to be found again in the next sampling period as a mixture of A and B strains (13/21 occurrences). Through the whole study, we did not see an infection convert between pure DWV-B and a double strain infection.

### 3.2. Proportions of Viral Load (in Genome Equivalent) of DWV-A and DWV-B

The relative viral load ratio of DWV-A and DWV-B in each colony (Appendix A) were compared in genome equivalents per bee (Figure 3). The proportion of DWV-B was constantly lower than DWV-A except for two colonies that were sampled in May-2019 which were DWV-A negative and two colonies from Jan-2018, which were negative for both DWV-A and B. The highest proportion of DWV-B load at the apiary level was in May-2019 which was 24%. In comparison, in May-2018, it was only 1%. At the apiary level, the DWV-A had the highest titer through this study. However, we noticed an increase in the proportion of the DWV-B load from 6% in 2018 to 14% in 2019.

### 3.3. Co-Infections and DWV Loads

The mean viral load (genome equivalents per bee) of DWV-A was compared between two infection conditions: single-variant infection and double-variant infection together with DWV-B (Table 1). In 2018, colonies infected with both DWV-A and DWV-B at the same time showed a significantly higher genome equivalent than colonies infected with DWV-A only (Wilcoxon Signed-Rank test, Z = −2.6, *p* = 0.0094). No significant difference was detected in 2019 (Z = −0.3, *p* = 0.76), although double-variant infected colonies had almost three times the viral load compared to single-variant infected colonies. Over the course of the two-year study, the viral load of DWV-A in double-variant infected colonies did not demonstrate a statistically significant higher genome equivalent compared to single-variant infected samples (Z = −1.5, *p* = 0.14).

### 3.4. Seasonal Effects on DWV Prevalence and Viral Loads in Colonies

The mean viral load of DWV-A and B was compared at the apiary level. Post-hoc analysis showed that the viral load levels of DWV-A (Mean = 2.9 × 10^8^, SE = 7.2 × 10^7^) were significantly higher than DWV-B (Mean = 0.75 × 10^8^, SE = 3.9 × 10^7^) overall (Wilcoxon Sign-Rank test Z = −4.4, *p* < 0.001). DWV-A viral load across different times of the year were relatively constant (Kruskal–Wallis Test, H = 3.5, *p* = 0.17) (Figure 4). In contrast, DWV-B had a significantly lower viral load in May compared to January (Wilcoxon Each Pair Z = −2.2, *p* = 0.03) and September (Z = −2.8, *p* = 0.006), respectively. When compared to DWV-A, the genome equivalent of DWV-B was only significantly lower in May (Z = −3.5, *p* = 0.0005). No significant difference between the viral load of DWV-A and B was shown in January (Z = −1.58, *p* = 0.11) and September (Z = −1.62, *p* = 0.10).

## 4. Discussion

The results of this two-year study on the prevalence of DWV variants on Oahu support the previous apiary level studies that also found an increased prevalence of DWV-B on varroa mite infected Hawaiian Islands [5,10,12]. In the previous study performed in Hawaii [10], researchers detected a higher proportion of DWV-B at the apiary level. In 2020, Brettell et al. [12] reported that the prevailing form of DWV-B was a recombinant closely associated with DWV-A. However, as we did not include sequencing, we are unable to identify recombinants between DWV-A and DWV-B. Consequently, it is possible that our results could have underestimated the actual prevalence of DWV-B in the apiary during the study period. Nonetheless, this does not alter the overall trend of an increased prevalence of DWV-B in Hawaii compared to ten years ago [10].

### 4.1. Seasonal Pattern of DWV Variants

At the apiary level, DWV-A exhibited a dominant status throughout the study compared to DWV-B, both on infection rate and viral load. Nevertheless, at a colony level, infection by DWV-B appeared to be intermittent. The detection of DWV-B was more likely to occur in May, when more than 70% of colonies tested positive for this strain in both years. Additionally, in 2019 two colonies tested positive for DWV-B only, for the first time without co-infection with DWV-A. When a colony tested positive for DWV-A during the initial sampling period it was likely to continue to have the DWV-A in the next sampling period which contributed to the overall greater prevalence of DWV-A in this study. In contrast, infections with only the DWV-B strain were rare and did not appear as persistent, as the affected colonies reverted to single DWV-A infection in the following sample period. Changes in strain prevalence in this study were associated with periods of colony growth and the concomitant reproductive opportunities for the varroa mites. During the transition from January to May more colonies changed status and tested positive for either strain A and B or only strain B compared to transitions between other periods suggesting a seasonal change linked to colony cycles.

Varroa mite levels and DWV loads are typically linked [2,5,13]. However, the relationship “mite, virus, and disease impact” is complex and variable. Among the factors that could potentially affect DWV levels are colony management [14], quantity of natural propolis [15], varroa mite levels [16], and climatic conditions [17,18,19]. In temperate regions, the prevalence and viral loads of DWV vary considerably throughout the year [17,18,20,21]. The impact of elevated viral loads can be linked to a seasonal susceptibility to disease. Steinmann et al. (2015) showed that winter bees had a reduced immunological response compared to summer bees, thus they were more susceptible to viral infections [20]. Unfortunately, the lack of data on varroa mite levels and the mild Hawaiian climate prevent us from examining possible correlations between a winter colony shrinkage and DWV load or variants in this study. However, according to Norton et al. (2021) [13], only the DWV-A strain showed a positive correlation with varroa mite levels. In contrast, recent studies have shown that only DWV-B was able to replicate in varroa mites which may suggests an increased prevalence of DWV-B in honeybee colonies [22,23].

Other factors that could potentially affect DWV levels are colony management and climate. The impact of mite treatments applied by the beekeeper and seasonality in colony growth have been considered as possibly playing a role in DWV infections. Varroa mite treatments are recommended to beekeepers to reduce the impact of DWV. However, beekeepers tend to treat according to their own schedules and with a diversity of products. Woodford et al. (2023) showed that varroa mite infestation was greatly reduced when mite treatments were coordinated across apiaries in a small island off Scotland [16]. The authors also tracked the shifts in DWV strain prevalence over the 3-year study and found that DWV-B became the dominant variant over this relatively short time span. In our study, the shift in prevalence between the A and B variants appears to be happening less rapidly, and DWV-A is still the dominant strain. It is unclear which factors are responsible for the increase in DWV-B prevalence observed by Woodford et al. (2023) [16], and whether the change was influenced by the movement of survivor colonies that harbored DWV-B or, whether it could have been a stochastic event [24]. On Oahu, beekeepers treat independently, and there is coordination among neighbors. Given the high density of small apiaries near our study site we can’t rule out that our colonies are part of a larger regional DWV population dynamic. However, according to Kevill et al. (2021) [22] the variant type of DWV does not appear to be related to miticide treatment and in the case of the colonies included in this study, they were all treated similarly.

The prevalence of DWV was different under different seasons in temperate areas with a higher DWV infection in either September in California [17] or spring in Europe [18]. Due to the warm and relatively stable tropical climate, honeybees are active year-round in Hawaii, and the viral loads and infection prevalence of DWV-A was relatively constant throughout the year. In comparison, DWV-B showed a higher infection rate at the apiary level but low viral load in May compared to January and September. Since May is one of the most active reproductive months for bees in this apiary, rather than environmental factors, colony level biological activities such as: high frequency of swarming, queen mating and high drone production rate [25,26] could influence the higher prevalence of DWV-B on Oahu.

### 4.2. Single Infection and Double Infection of DWV A and B

The dynamics between DWV-A and B strains is not yet fully understood; differential mortality between the variants and the possible epidemiological impacts of single versus recombinant infections are still being discussed [4,27].

Earlier studies suggested that the different strains of DWV were involved in an evolutionary competition which could lead to the establishment of a dominant variant, the Superinfection Exclusion (SIE) of DWV strains. Subsequent studies on SIE revealed that DWV-A failed to persist in colonies with an increased prevalence of DWV-B [28,29]. The A and B variants of DWV were predicted to outcompete each other in a colony depending on sequence of arrival [28] and furthermore, it appears that the competition between DWV-A and B is fleeting, as DWV-B has been shown to have a higher replication efficiency than DWV-A in varroa mites [30]. However, in our study, results clearly showed how a colony can test positive for a single DWV strain only to reverse to a different one in the next sampling period and colonies frequently tested positive for both strains during our study indicating simultaneous infections. The sequential changes and reversal in strain dominance in DWV we observed may be due to a few factors, including the genetic differences between A and B strains in Hawaii, coupled with the multiple routes of virus acquisition by bees within the hive, including food resources, vertical, horizontal, and finally, varroa mite replication and transmission [13,25,31].

The results of this study also show that even in the relatively thermally stable subtropical climate there were seasonal trends to the DWV infection and that viral load levels differed between colonies. Consequently, it seems that biotic factors such as colony health, mite levels, food resource quality and abundance, may play a role in the dynamics of viral strains in climates where a distinct winter and summer seasonality is not even observed. Similarly, significant differences in infection patterns of DWV A and B have been shown to exist not only across colonies, but also between bees of different patrilines in the same colony [32]. Individual bee susceptibility to diseases, whether viral or bacterial, and increased worker mortality can lead to colony levels effects, which makes it difficult to predict the trajectory of a colony without more long-term studies [33]. More data is needed to understand how the genetic make-up of the bee host, the mite, and the diseases interact.

## Figures and Tables

**Figure 1 insects-15-00219-f001:**
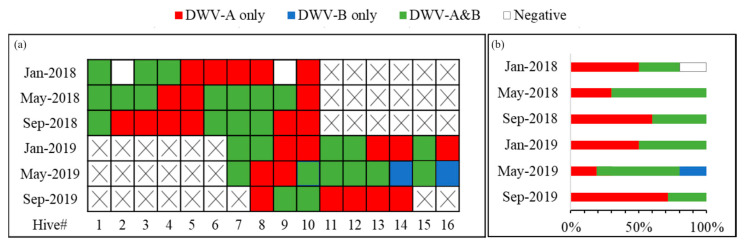
(**a**) The chart shows the DWV status of each of the 16 colonies in the study (Hive# on the *X* axis) across the different sampling periods in 2018 and 2019 (*Y*-axis). Colonies #7 through #10 were sampled across both years. The DWV status of the colonies are color coded as follows: a red square indicates a hive that was only positive for DWV-A, blue indicates type B only positive, and green indicates the colony was infected with both DWV-A and B. White indicates no detection of either variant. A cross in the chart indicates that the colony was not sampled at that time. (**b**) The bar graph shows the proportion of colonies with different DWV variants (*X*-axis) during each sampling period. The percentage on the *X*-axis indicates the prevalence of each different type of infection during that sample period.

**Figure 2 insects-15-00219-f002:**
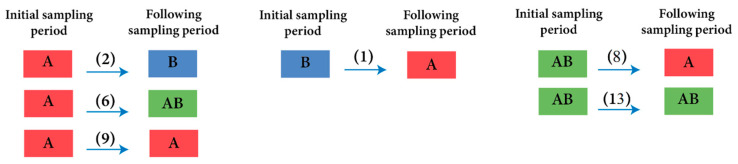
Illustrates the observed shifts in DWV variants in individual colonies from one sampling period to the next. DWV strains are coded by letters A, B, and double infection AB. Colonies are represented by rectangular boxes color coded to represent their DWV strain status. Arrows point to the direction of the change and the numbers in parentheses represent the number of occurrences the shift was observed in.

**Figure 3 insects-15-00219-f003:**
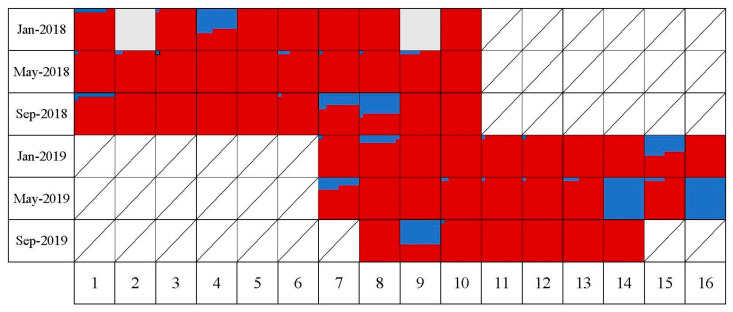
This chart shows the proportions of DWV-A (red) and DWV-B (blue) in each individual hive across the study period. Each square represents a unique sampling period for each colony and the relative ratio of the genome equivalents of DWV-A and DWV-B is illustrated by the proportional area of different colors in each square. Consequently, a solid red or blue square represents a 100% single infection with DWV-A and DWV-B, respectively, and a solid white square is a 100% negative sample.

**Figure 4 insects-15-00219-f004:**
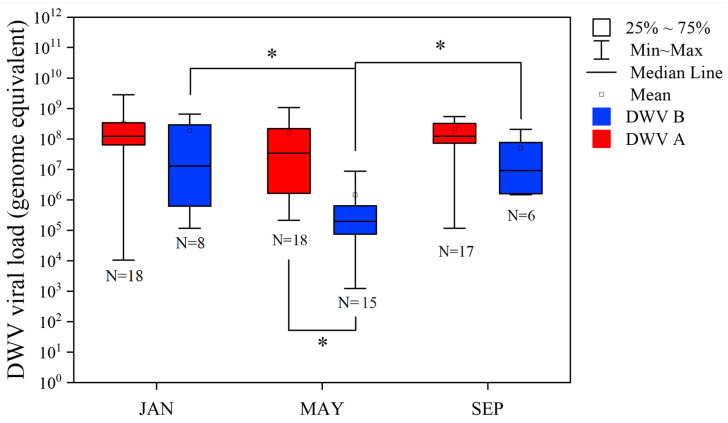
These boxplots show the viral load (in genome equivalents per bee) between DWV-A and DWV-B at the apiary level (mean, median, percentiles, and range). The asterisks indicate a significant difference (Wilcoxon Signed-Rank test, *p* < 0.05) between linked months.

**Table 1 insects-15-00219-t001:** At the apiary level, the mean DWV-A viral load comparison between two different infection conditions over two years (N = sample size).

Genome Equivalents of DWV-A (* 10^8^ Gene Copies/Bee)				
	Single Variant Infection (Only A)	N	Double Variant Infection (DWV-A + DWV-B)	N
2018 *	1.00	14	3.35	14
2019	2.53	12	4.88	13
Total	1.7	26	4.09	27

* 2018 row: significant difference between single- and double-variant infection (Wilcoxon Signed-Rank test, *p* < 0.05).

## Data Availability

Data is contained within the article.

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
