# Peer review of "Seasonal Variability in the Prevalence of DWV Strains in Individual Colonies of European Honeybees in Hawaii"

_insects, 2024, doi:10.3390/insects15040219_

Round 1

Reviewer 1 Report (New Reviewer)

Comments and Suggestions for Authors

DWV is the most important virus associated with varroa mite parasitism of honey bees and the likely most important cause of colony death. In temperate regions, DWV variant B is taking over from variant A. This short paper presents data on the change in the variant of DWV (A and B) across time using a small number of colonies on Hawaii, demonstrating that DWV-B has not taken over from DWV-A.

Major Comment (numbered and by line number, where relevant)

1. Change italicized ‘Varroa’ to non-italicized ‘varroa’ throughout and add the species name (Varroa destructor) on first mention of ‘varroa’ (if journal style is to conform to norms of the English language). 

2. 113 3Ct is a large difference. Most studies use a threshold difference of 1 unit per duplicate qPCR as a quality threshold. The correct term is quantification cycle (Cq).

3. 296-298 This sentence is illogical, or at least poorly formulated. How does ‘fleeting competition between variants A and B’ of the first part of the sentence relate to the second part of the sentence, that ‘DWV-B replicates in varroa mites’? If the authors wish to suggest that DWV-B should therefore take over, then their data are not in accord with the suggestion. 

4. State when DWV-B was first detected on Hawaii.

Comments on the Quality of English Language

Minor Comments (by line number)

14. Delete ‘are present’ as otherwise the sentence is not grammatically correct.

21 Delete ‘And’ and use ‘variation’ (the singular form).

51 Change ‘to be’ to ‘become’ as otherwise the language is not correct.

58 Language: ‘the’ is required before ‘Varroa’. `Alternatively, make ‘mite’ plural (‘mites’). 

81 Misspelling: change ‘identify’ to ‘identity’.

88 Add ‘an’ before ‘RNeasy’ and ‘the’ before ‘manufacturer’s’.

96 Delete ‘, respectively,’.

107 Add ‘an’ before ‘R2’.

112 Add ‘in’ before ‘duplicate’

113 Change ‘3 Ct values difference were redone to achieve the accurate answer.’ to ‘3 Ct value difference were reanalyzed’. You could add here the criterion you used to decide what was accurate and how you then chose the Ct value (e.g. mean of values within 3 Ct).

119 Add ‘and’ before ‘consequently’. Alternatively, use a full stop or semi-colony in place of ‘and’.

121 Change ‘test’ to ‘tests’.

123 ‘Each’ should possibly be ‘matched pair’.

174 Change ‘two colonies from Jan-2018 were detected negative for both DWV-A and B’ to ‘two colonies from Jan-2018, which were negative for both DWV-A and B’

176-177 Change ‘At the apiary level, the genome equivalent of DWV-A was the prevalent strain through this study’ to ‘At the apiary level, the DWV-A had the highest titer through this study’

221 Add ‘the’ before ‘apiary’.

232 Change ‘more’ to ‘greater than’.

238 ‘ingle’ ïƒ  ‘single’.

261-2 Change ‘producers’ to ‘beekeepers’.

288 Change the comma to a semi-colon.

Author Response

Major Comment (numbered and by line number, where relevant)

  1. Change italicized ‘Varroa’ to non-italicized ‘varroa’ throughout and add the species name (Varroa destructor) on first mention of ‘varroa’ (if journal style is to conform to norms of the English language). 

Corrections were made.

  1. 113 3Ct is a large difference. Most studies use a threshold difference of 1 unit per duplicate qPCR as a quality threshold. The correct term is quantification cycle (Cq).

Response: Thanks for the correction. We used the analysis protocol from Kevill et. al. 2017 of a 3 Ct value deviation as a threshold.  In this study, the majority of the  samples had Ct values difference1.  Triplicate samples and Cq value differences of more than 1 will be considered in future studies involving qRT-PCR.

  1. 296-298 This sentence is illogical, or at least poorly formulated. How does ‘fleeting competition between variants A and B’ of the first part of the sentence relate to the second part of the sentence, that ‘DWV-B replicates in varroa mites’? If the authors wish to suggest that DWV-B should therefore take over, then their data are not in accord with the suggestion. 

            Minor rewording to the paragraph to clarify our statement. Line 296.

  1. State when DWV-B was first detected on Hawaii.

            This info was added in the introduction section. Line 53.

Minor Comments (by line number)

  1. Delete ‘are present’ as otherwise the sentence is not grammatically correct.

            Correction was made.

21 Delete ‘And’ and use ‘variation’ (the singular form).

            Correction was made.

51 Change ‘to be’ to ‘become’ as otherwise the language is not correct.

            Correction was made.

58 Language: ‘the’ is required before ‘Varroa’. `Alternatively, make ‘mite’ plural (‘mites’). 

            Correction was made.

81 Misspelling: change ‘identify’ to ‘identity’.

            Correction was made.

88 Add ‘an’ before ‘RNeasy’ and ‘the’ before ‘manufacturer’s’.

            Correction was made.

96 Delete ‘, respectively,’.

            Correction was made.

107 Add ‘an’ before ‘R2’.

            Correction was made.

112 Add ‘in’ before ‘duplicate’

            Correction was made.

113 Change ‘3 Ct values difference were redone to achieve the accurate answer.’ to ‘3 Ct value difference were reanalyzed’. You could add here the criterion you used to decide what was accurate and how you then chose the Ct value (e.g. mean of values within 3 Ct).

            Sentence was rewritten to include how the Ct values were selected and calculated.

119 Add ‘and’ before ‘consequently’. Alternatively, use a full stop or semi-colony in place of ‘and’.

            Correction was made.

121 Change ‘test’ to ‘tests’.

            Correction was made.

123 ‘Each’ should possibly be ‘matched pair’.

            This sentence was changed to: ‘…were compared for each pair using Wilcoxon (non-parametric)’

174 Change ‘two colonies from Jan-2018 were detected negative for both DWV-A and B’ to ‘two colonies from Jan-2018, which were negative for both DWV-A and B’

            Correction was made.

176-177 Change ‘At the apiary level, the genome equivalent of DWV-A was the prevalent strain through this study’ to ‘At the apiary level, the DWV-A had the highest titer through this study’

            Correction was made.

221 Add ‘the’ before ‘apiary’.

            Correction was made. ‘the’ was added before ‘previous’.

232 Change ‘more’ to ‘greater than’.

            We opted not to change based on the comment since "more likely" is suitable in this context.

238 ‘ingle’ à ‘single’.

            Correction was made.

261-2 Change ‘producers’ to ‘beekeepers’.

            Correction was made.

288 Change the comma to a semi-colon.

            Correction was made.

Reviewer 2 Report (Previous Reviewer 1)

Comments and Suggestions for Authors

This manuscript describes the prevalence and viral load of both DWV-A and DWV-B in managed honey bee hives in Hawaii across several sampling time points in 2018 and 2019. 

Overall, this revised version of the manuscript is well written and clear. The results and conclusions do not provide any great insights into our understanding of the relationship between the two master variants of DWV. However, these longitudinal studies are useful for monitoring the prevalence of DWV-A and DWV-B across time and space and help us determine how these two viral strains interact in different environments and at different time points.

Any previous concerns about the presentation of the methods and results have been resolved in the updated version. I only have one substantive comment and that is to inquire how the cDNA reactions were standardized to 50ng/uL (page 3, line 95). This cannot be accomplished using spectrophotometry (Nanodrop), since the dNTPs and primers in the reaction will also absorb UV light at 260nM. I was wondering if the authors used another method, such as Qubit, to standardize their concentrations. If so, including that detail in the manuscript would be helpful. 

I look forward to seeing the final manuscript in print.

Author Response

I only have one substantive comment and that is to inquire how the cDNA reactions were standardized to 50ng/uL (page 3, line 95). This cannot be accomplished using spectrophotometry (Nanodrop), since the dNTPs and primers in the reaction will also absorb UV light at 260nM. I was wondering if the authors used another method, such as Qubit, to standardize their concentrations. If so, including that detail in the manuscript would be helpful. 

            Yes, we used Qubit to standardize cDNA samples. This is added to the manuscript (line 95). Thanks for clarifying this issue.

This manuscript is a resubmission of an earlier submission. The following is a list of the peer review reports and author responses from that submission.

Round 1

Reviewer 1 Report

Comments and Suggestions for Authors

This manuscript describes the prevalence and viral load of both DWV-A and DWV-B in managed honey bee hives in Hawaii across several sampling time points in 2018 and 2019.  

Overall, this manuscript is fairly well written and clear. The results and conclusions do not provide any great insights into our understanding of the relationship between the two master variants of DWV. But these longitudinal monitoring studies are useful and allow comparisons between how DWV-A and DWV-B interact in different environments and at different time points.

Major Concerns:

1.     The authors use the published and validated primer sets from Kevill et al (2017) to measure levels of both DWV-A and DWV-B. These primers only provide information about the sequence of the genome at one location – primarily near the 3’ end within the RNA-dependent RNA polymerase gene. These primer sets do not allow the researchers to identify any recombinant viral genomes that would contain regions from both DWV-A and DWV-B and may be underreporting the level of DWV-B sequences found in these hives.

While it is not feasible for the authors to carry out RNA-Seq to analyze the level of recombinants in their samples, they should at least acknowledge the presence of recombinant viral genomes in these hives and how it may alter their interpretations of their results (see reference 15, Brettell et al, Viruses, 2020).

2.     I have some concerns about using only duplicate samples for qRT-PCR as it is considered more accurate to use triplicate samples. The authors do state that samples that were
3 Ct values different were rerun – but that represents and 8-fold difference in absolute genome equivalents. I would have been more comfortable with any samples that were more than 1 full Ct value different being rerun. While it is not feasible for the samples to be reanalyzed, the authors should keep this in mind for future projects.

3.     In the Materials and Methods, the authors do not adequately explain what was used as the positive control (plasmids containing cloned amplicons I assume) and how the standard curves were carried out.

Minor concerns and suggestions for improvement:

1.     While in general this manuscript is well written, there are several locations where minor editing is required to improve the readability.

·      Line 14 – 15: The authors mention 4 master strains but only describe three of them. If the fourth strain is not currently circulating – is it worth it to even mention it?

·      Line 100: “105” should probably be 10^5 genome copies

·      Line 109-110: do not need to describe how to calculate copy numbers from mass and size of a plasmid. That is routine knowledge.

·      Lines 122-124: remove text from template.

·      Lines 201-202: either use “are similar to” or “aligns with” but not both.

·      Lines 254-255: this sentence is unclearly worded. The formation of recombinant strains is not due to RNA interference, it is the outcome of the competition between the master variants that is due to RNA interference. Rewrite this sentence to be more accurate and clearer.

2.     The entire Discussion could be better organized. The authors may make some good points about differences between the Hawaiian environment and other locations as well as the potential competition between the two variant genomes. But overall, it is not as focused or as clear as it could be.

Author Response

  1. The authors use the published and validated primer sets from Kevill et al (2017) to measure levels of both DWV-A and DWV-B. These primers only provide information about the sequence of the genome at one location – primarily near the 3’ end within the RNA-dependent RNA polymerase gene. These primer sets do not allow the researchers to identify any recombinant viral genomes that would contain regions from both DWV-A and DWV-B and may be underreporting the level of DWV-B sequences found in these hives.

While it is not feasible for the authors to carry out RNA-Seq to analyze the level of recombinants in their samples, they should at least acknowledge the presence of recombinant viral genomes in these hives and how it may alter their interpretations of their results (see reference 15, Brettell et al, Viruses, 2020).

Response: We revised the first paragraph in the discussion section.

  1. I have some concerns about using only duplicate samples for qRT-PCR as it is considered more accurate to use triplicate samples. The authors do state that samples that were 3 Ct values different were rerun – but that represents and 8-fold difference in absolute genome equivalents. I would have been more comfortable with any samples that were more than 1 full Ct value different being rerun. While it is not feasible for the samples to be reanalyzed, the authors should keep this in mind for future projects.

Response: We agree it is more accurate to use triplicate samples for qRT-PCR, however previous studies have used duplicate samples in their analysis of viral load (Kandel et. al., 2023, Norton et. al. 2021, Dosch et. al. 2021).

We used the analysis protocol from Kevill et. al. 2017 of a 3 Ct value deviation as a threshold.  In this study, 93.3% of the total samples had a Ct value difference ≤1.  Triplicate samples and Ct value differences of more than 1 will be considered in future studies involving qRT-PCR.

  1. In the Materials and Methods, the authors do not adequately explain what was used as the positive control (plasmids containing cloned amplicons I assume) and how the standard curves were carried out.

Response: We added an explanation of that in the method section.

  1. While in general this manuscript is well written, there are several locations where minor editing is required to improve the readability.

  • Line 14 – 15: The authors mention 4 master strains but only describe three of them. If the fourth strain is not currently circulating – is it worth it to even mention it?

Response: We have removed line the sentence describing DWV-C and D from the manuscript and their related citations.  

  • Line 100: “105” should probably be 10^5 genome copies

Response: revised in the manuscript.

  • Line 109-110: do not need to describe how to calculate copy numbers from mass and size of a plasmid. That is routine knowledge.

Response: We removed this equation from the manuscript.

  • Lines 122-124: remove text from template.

Response: revised in the manuscript.

  • Lines 201-202: either use “are similar to” or “aligns with” but not both.

Response: sentences were revised in the manuscript.

  • Lines 254-255: this sentence is unclearly worded. The formation of recombinant strains is not due to RNA interference, it is the outcome of the competition between the master variants that is due to RNA interference. Rewrite this sentence to be more accurate and clearer.

Response: The paragraph was shortened, and this sentence was deleted.

  1. The entire Discussion could be better organized. The authors may make some good points about differences between the Hawaiian environment and other locations as well as the potential competition between the two variant genomes. But overall, it is not as focused or as clear as it could be.

Response: discussion section was revised in the manuscript.

Reviewer 2 Report

Comments and Suggestions for Authors

In this study, the authors present an examination of DWV strain variability and changes within hives belonging to a single apiary on the Island of Oahu, Hawaii. Standard beekeeping and molecular biology methods were utilised to obtaining data providing insight into patterns of presence of two major DWV strains, DWV-A and DWV-B.

Overall, a well written paper which displays viral dynamics within a semi-controlled environment. Whilst successful in showing certain aspects of viral dynamics, the short timeframe and low number of usable datapoints across seasons represent more a proof-of-concept study rather than a successful examination of persistence across any meaningful timeframe. Additionally, it is understood that viral levels vary wildly even at the individual honeybee level meaning whole hive observations may mask any differences within colonies. Is there any data examining within colony differences in viral load? Saying that, the manuscript is still beneficial in that it provides some insight into the dynamics of a major pathogen of honeybees.

Make clearer how the introduction of replacement colonies is acceptable and comparable to data obtained from surviving colonies.

Preamble and introduction

The introduction for the most part is relevant and well written. More details could be provided in places, providing better context to the study and what it’s trying to achieve/ show.

Line 2: Describing the study as examining persistence may be a bit strong. The study took place over 2 years with only 4 colonies surviving across both years, and 3 colonies being sampled at all collection points. Consider rephrasing.

Line 37: double space

Line 43: Provide Latin name for European honeybee

Line 44, 61 etc: Italicize the Latin name, Varroa destructor. Find consensus in terminology for parasite throughout manuscript.

Line 49: Mention of DWV-C and DWV-D, yet not examined in this study. May not be relevant in introduction unless more context to inclusion provided.

Line 67: Study performed in a single apiary. How many other apiaries on the island? What is their proximity to the test apiary?

Methods

Was there any testing for other pathogens within the colonies?

Line 78-80: Explain why hives were treated for Varroa (e.g. to remove confounding effect of parasite in viral strain selection).

Line 101: Provide primer oligo sequences. Whilst reference is provided it’s not clear what primers from the reference were used.

Line 110: It’s not clear if you used the plasmids from the above reference. Provide details, including dilutions used and source.

Line 110: Consider further normalising to a “per bee” level, or make clear in results that viral loads are “per hive”.

Results

Line 122: Statement of subsections isn’t required.

Line 126: “All colonies” better replaced with “all tested colonies” due to introduction of new colonies later in study.

Line 128: “Infection rate” used when preexisting infection in colonies was tested. Consider changing to “prevalence rate”.

Figure 1 and 3: Blue and grey colours difficult to differentiate.

Figure 2: It isn’t clear what you’re trying to show here. Describe further in text or consider omitting/ adapting.

Line 162-163: Rewrite to make clearer.

Line 180: Avoid use of “borderline significantly”

Any comparison of colonies that survived the full 2 years vs early leavers vs late introductions?

Discussion

Reformat subheadings to be as in the methods. Currently unclear as not italicized.

Line 225-227: Expand how all hives were treated similarly. Introduced hives were transported whilst those already on-site were established. This may have influenced behaviour of the bees, for instance making them more protective of their colony, lowering the risk of DWV transmission by robbing bees. Similarly, try to better address the fact that certain colonies had to be removed from the study. Why were they removed? Was this due to, or would it impact, viral dynamics within the colony? As such is this data comparable with the surviving colonies? Make clearer to help the reader trust the data more.

Other similar studies occurred in similar isolated systems in other climates such as that of Woodford et al (2023) (https://doi.org/10.3390/v9110314). Consider comparing with these to put your research in the wider context of honeybee health. Make more of a discussion on why your data is important and how it can be used.

Comments on the Quality of English Language

The manuscript, for the most part, is clearly written. It would however benefit from rewriting in places to improve clarity (e.g. Line 54: “Although DWV-B has been not to be dominant..”).

Author Response

In this study, the authors present an examination of DWV strain variability and changes within hives belonging to a single apiary on the Island of Oahu, Hawaii. Standard beekeeping and molecular biology methods were utilised to obtaining data providing insight into patterns of presence of two major DWV strains, DWV-A and DWV-B.

Overall, a well written paper which displays viral dynamics within a semi-controlled environment. Whilst successful in showing certain aspects of viral dynamics, the short timeframe and low number of usable datapoints across seasons represent more a proof-of-concept study rather than a successful examination of persistence across any meaningful timeframe. Additionally, it is understood that viral levels vary wildly even at the individual honeybee level meaning whole hive observations may mask any differences within colonies. Is there any data examining within colony differences in viral load? Saying that, the manuscript is still beneficial in that it provides some insight into the dynamics of a major pathogen of honeybees.

Response:  In this study, we did not conduct a viral load analysis within a colony.

Make clearer how the introduction of replacement colonies is acceptable and comparable to data obtained from surviving colonies.

Response: The six colonies that were added to this study in 2019 were selected by the same standard as we selected hives in 2018. All the colonies were located in the same apiary. We have revised the sentence in the methods section and try to make this clearer.  

Preamble and introduction

The introduction for the most part is relevant and well written. More details could be provided in places, providing better context to the study and what it’s trying to achieve/ show.

Line 2: Describing the study as examining persistence may be a bit strong. The study took place over 2 years with only 4 colonies surviving across both years, and 3 colonies being sampled at all collection points. Consider rephrasing.

Response: We decided to remove persistence from the title.

Line 37: double space

Response: fixed in the manuscript.

Line 43: Provide Latin name for European honeybee

Response: added in the manuscript.

Line 44, 61 etc: Italicize the Latin name, Varroa destructor. Find consensus in terminology for parasite throughout manuscript.

Response: revised in manuscript.

Line 49: Mention of DWV-C and DWV-D, yet not examined in this study. May not be relevant in introduction unless more context to inclusion provided.

Response: We have removed line the sentence describing DWV-C and D from the manuscript and their related citations. 

Line 67: Study performed in a single apiary. How many other apiaries on the island? What is their proximity to the test apiary?

Response: Within 3km of the study apiary are several other known apiaries, but none within 1 km.

Methods

Was there any testing for other pathogens within the colonies?

Response: No, for this study, we only focused on DWV.

Line 78-80: Explain why hives were treated for Varroa (e.g. to remove confounding effect of parasite in viral strain selection).

Response: Colonies were maintained using local beekeeping practices. During this study, colonies had Varroa treatment once a year.    

Line 101: Provide primer oligo sequences. Whilst reference is provided it’s not clear what primers from the reference were used.

Response: Primer sequences were added in the manuscript. 

Line 110: It’s not clear if you used the plasmids from the above reference. Provide details, including dilutions used and source.

Response: Templates used in this calculation were purified amplicons from known positive samples. More information was added in manuscript section 2.3.

Line 110: Consider further normalising to a “per bee” level, or make clear in results that viral loads are “per hive”.

Response: Edits were made for better understanding the measure of viral load in method section and Table 1. In this study, viral load is expressed as genome equivalent per bee which indicates the gene copies of DWV per bee.

Results

Line 122: Statement of subsections isn’t required.

Response: Deleted from the manuscript.

Line 126: “All colonies” better replaced with “all tested colonies” due to introduction of new colonies later in study.

Response: Revised in the manuscript.

Line 128: “Infection rate” used when preexisting infection in colonies was tested. Consider changing to “prevalence rate”.

Response: Revised in the manuscript.

Figure 1 and 3: Blue and grey colours difficult to differentiate.

Response: We changed the grey color to white. Figures were corrected in the manuscript.

Figure 2: It isn’t clear what you’re trying to show here. Describe further in text or consider omitting/ adapting.

Response: We revised the figure and its legend, and we added a new description in the manuscription. A paragraph was also added in the discussion on the result of Figure 2.

Line 162-163: Rewrite to make clearer.

Response: Sentence is revised.

Line 180: Avoid use of “borderline significantly”

Response: We deleted it from the manuscript.

Any comparison of colonies that survived the full 2 years vs early leavers vs late introductions?

Response: No, we did not make any comparisons because these excluded colonies from 2018 were not suitable for proper diagnostic testing.

Discussion

Reformat subheadings to be as in the methods. Currently unclear as not italicized.

Response: Revised in the manuscript.

Line 225-227: Expand how all hives were treated similarly. Introduced hives were transported whilst those already on-site were established. This may have influenced behaviour of the bees, for instance making them more protective of their colony, lowering the risk of DWV transmission by robbing bees. Similarly, try to better address the fact that certain colonies had to be removed from the study. Why were they removed? Was this due to, or would it impact, viral dynamics within the colony? As such is this data comparable with the surviving colonies? Make clearer to help the reader trust the data more.

Response: Overall, colonies that were two small after the “2018-19 winter” were excluded from this study because of their very small colony size due to failed requeening. None of them were died from disease. These weak colonies stayed at the same spots and some other healthy colonies (which were not included in 2018) were included for the 2019 data collection. Changes were made in the methods section to make this clearer.

Other similar studies occurred in similar isolated systems in other climates such as that of Woodford et al (2023) (https://doi.org/10.3390/v9110314). Consider comparing with these to put your research in the wider context of honeybee health. Make more of a discussion on why your data is important and how it can be used.

Response: we have included this article and added a paragraph in the discussion section.